# Improved Light and In Vitro Digestive Stability of Lutein-Loaded Nanoparticles Based on Soy Protein Hydrolysates via Pepsin

**DOI:** 10.3390/foods11223635

**Published:** 2022-11-14

**Authors:** Renyi Wu, Xuejiao Qie, Zhaojun Wang, Qiuming Chen, Maomao Zeng, Jie Chen, Fang Qin, Zhiyong He

**Affiliations:** 1State Key Laboratory of Food Science and Technology, Jiangnan University, Wuxi 214122, China; 2International Joint Laboratory on Food Safety, Jiangnan University, Wuxi 214122, China

**Keywords:** lutein, soy protein hydrolysates, encapsulation, nanoparticles, stability

## Abstract

In order to improve the water solubility and stability of lutein, soy protein isolates (SPI) and their hydrolysates via pepsin (PSPI) and alcalase (ASPI) were used as nanocarriers for lutein to fabricate the lutein-loaded nanoparticles (LNPS) of SPI, PSPI, and ASPI. The encapsulation properties, light, and in vitro digestive stability of lutein in nanoparticles, and protein–lutein interactions were investigated. Compared with SPI-LNPS and ASPI-LNPS, PSPI-LNPS was characterized by uniform morphology (approximately 115 nm) with a lower polydispersity index (approximately 0.11) and higher lutein loading capacity (17.96 μg/mg protein). In addition, PSPI-LNPS presented the higher lutein retention rate after light exposure (85.05%) and simulated digestion (77.73%) than the unencapsulated lutein and SPI-LNPS. Fluorescence spectroscopy revealed that PSPI had stronger hydrophobic interaction with lutein than SPI, which positively correlated with their beneficial effects on the light and digestive stability of lutein. This study demonstrated that PSPI possessed significant potential for lutein delivery.

## 1. Introduction

Lutein, a type of natural carotenoid found in vegetables, fruits, and eggs [1], is beneficial for a variety of health problems, most notably photo-oxidative retinal damage, eye inflammation, neurological disorders, and oral diseases [2]. However, lutein’s rapid photodegradation and low aqueous solubility make it difficult to apply and absorb [3]. In order to overcome these obstacles, it has been widely confirmed that developing nutrient delivery systems is a promising solution. Proteins, polysaccharides, and starches derived from food are frequently used for nanoencapsulation.

Emulsion systems were widely used to encapsulate carotenoids. However, fabricating lutein-loaded nanoemulsions caused a low encapsulation efficiency due to the homogenization process [4]. Fabricating fat-free nanoparticles is another effective method [5]. In recent years, several proteins have been used to encapsulate lutein by an antisolvent method. Glycosylated casein and whey protein were used to create pH and ionic stable lutein-loaded nanoparticles [6,7]. Zein and its hydrolysates were used to encapsulate lutein [8]. Zein-derived peptides exhibited significantly increased lutein solubility and stability against in vitro digestion compared with zein. Additionally, rice protein hydrolysates were also successfully used to fabricate lutein-loaded nanoparticles [9]. As natural and environmentally friendly food components, plant proteins have several beneficial effects on physical health by lowering the risk of cardiovascular disease, diabetes, and certain types of cancer [10]. However, as a widespread source of plant protein, soy protein has not been shown to encapsulate lutein.

SPI has a much higher molecular weight than animal protein, and the hydrophobic groups are mostly embedded. Due to their high molecular weight and limited hydrophobic binding sites on the surface, hydrophobic nutrients have a lower loading capacity in nanoparticles than animal proteins [11]. Through physical or chemical modification of SPI, additional hydrophobic groups can be exposed to combine with hydrophobic nutrients, thereby increasing the loading capacity of hydrophobic nutrients in SPI. For example, heat treatment, ultrasonic treatment, the addition of mild denaturants, and enzymatic hydrolysis can all be used to dissociate and reassemble SPI [12,13,14]. Enzymatic modification is a hotspot for plant protein structure modification research due to its mild production conditions, high safety, high specificity, and ease of control. Alcalase and pepsin were used to prepare soy protein hydrolysates for curcumin encapsulation via sonoassembled and pH-driven methods [15,16]. These methods were based on the solubility of curcumin in an alkali solution to trigger reassembling with protein. Curcumin belongs to phenolic compounds, while lutein is a type of carotenoid. The hydrophobicity of lutein is higher than curcumin. It could not dissolve in both acid and alkali solutions, so these methods could not be applied to lutein. The effectiveness of soy protein hydrolysates for lutein encapsulation still needs to be verified. It has been demonstrated that the nanodelivery of carotenoids such as beta-carotene and lutein by the antisolvent method via peptides derived from other proteins improves their stability, solubility, and bioactivity when compared to the original proteins [8,17]. Based on these findings, we hypothesize that certain SPI-derived peptides may be more effective at encapsulating lutein than SPI.

This study sought to increase lutein’s encapsulation effect via enzymatic hydrolysis of SPI, thereby increasing lutein’s light and in vitro digestive stability. First, SPI hydrolysates were prepared using pepsin (PSPI) and alcalase (ASPI), and their structural properties and lutein encapsulation properties were analyzed. The lutein-loaded SPI hydrolysates nanoparticles with the best encapsulation properties and particle characteristics were chosen for comparison to unencapsulated lutein and lutein-loaded SPI nanoparticles in terms of stability against light and in vitro digestion. Meanwhile, the surface hydrophobicity, fluorescence spectroscopy, circular dichroism spectrum, Fourier transform infrared spectrum, and X-ray diffraction analysis were used to investigate the potential mechanism leading to the different encapsulation effects and lutein stability when SPI and its hydrolysates nanoparticles were used. This work will aid in the improvement of lutein’s aqueous solubility, stability, and bioavailability, and the development of lutein-enriched nutraceuticals.

## 2. Materials and Methods

### 2.1. Materials

Lutein (purity > 90%), alcalase (enzyme activity > 200 U/mg), bile salts, and molecular weight standards including bacitracin, aprotinin, myohemoglobin, bovine serum albumin, and thyroglobulin were purchased from Shanghai Yuanye Biotechnology Co., Ltd. (Shanghai, China). Soybeans were bought from Fengyuan Zhongye Co., Ltd. (Lianyungang, China). Pepsin (enzyme activity > 2500 U/mg), trypsin (enzyme activity > 2500 U/mg), 8-aniline-1-naphthalensulfonic acid (ANS), and trinitrobenzenesulfonic acid (TNBS) were purchased from Sigma-Aldrich Co., (St. Louis, MO, USA). All other reagents and solvents were of at least analytical grade.

### 2.2. Preparation of SPI and Its Hydrolysates by Pepsin and Alcalase (PSPI and ASPI)

SPI was prepared from soybeans with reference to a previous publication [18]. First, soybeans were pulverized and defatted with a threefold concentration of n-hexane:ethyl alcohol (9:1, *v*/*v*). After drying the organic solvent in the air for 12 h, the defatted soy flour was blended with deionized water at a 1:10 (*w*/*v*) ratio with the pH adjusted to 8.0 using 2 M NaOH. The dispersion was stirred for 2 h at a constant pH of 8.0 and centrifuged for 20 min at 10,000× *g*, and the supernatant was adjusted to pH 4.5 with 2 M HCl and centrifuged at 3300× *g* for 20 min (Xiangyi Appliance Co., Ltd., Beijing, China) The precipitate was redissolved in deionized water at a ratio of 1:4 (*w*/*v*) for at least 3 h while maintaining a pH of 7.0. The solution was lyophilized to yield SPI with a protein content of 90.23%, as determined by the Kjeldahl method. The SPI was stored at −80°C prior to use.

PSPI and ASPI were prepared in a manner similar to that used by other researchers [15,19]. SPI was completely dissolved in deionized water (5%, *w*/*v*) after overnight hydration. Enzymatic hydrolysis both took 45 min using pepsin (pH 2.0, 37 °C) and alcalase (pH 8.0, 55 °C), respectively. Enzyme activity was 2500 U/g protein for pepsin and 2000 U/g protein for alcalase. After adjusting pH to 7.0, the solution was boiled for 10 min at 100 °C in a water bath to deactivate enzymes and then rapidly cooled. After centrifuging the solution at 8000× *g* for 15 min, the supernatant was collected and lyophilized to yield PSPI and ASPI. PSPI and ASPI were stored at −80°C prior to use.

### 2.3. Characteristics of SPI, PSPI, and ASPI

#### 2.3.1. Degree of Hydrolysis (DH) Measurement

The DH of PSPI and ASPI was determined using the TNBS assay described by Adler-Nissen [20]. Specifically, 0.125 mL protein solutions (1 mg/mL) were thoroughly mixed with 1 mL phosphate buffer (pH 8.2, 0.2 M) and 1 mL TNBS (0.1%). After 1 h of dark reaction at 50 °C, 2 mL of 0.1 M HCl was added to terminate the reaction. The absorbance at 340 nm was measured until the solution reached room temperature. Additionally, the unhydrolyzed SPI and the standard L-leucine (0–2.5 mmol/L) were used to react together. The following Equation (1) was used to determine the DH:DH = h/h_tot_ × 100%(1)
where h is the concentration of broken peptide bonds (mmol/g), and h_tot_ is the total concentration of broken peptide bonds after thorough hydrolysis, which is 7.8 mmol/g for soy protein.

#### 2.3.2. Molecular Weight (MW) Distribution

The molecular weight distribution of SPI and its hydrolysates was determined using an HPLC system (Waters 2695, Milford, MA, USA) equipped with a size exclusion chromatography column according to the method described by Jiang et al. [21]. The testing condition was listed below.

Chromatography column: protein KW-804 (8 × 300 mm, 5 μm, Shodex Co., Tokyo, Japan); flow rate: 1 mL/min; wavelength: 220 nm; column temperature: 25 °C; mobile phase: 50 mM phosphate buffer (pH 7.0) containing 30 mM sodium chloride; injection volume: 10 μL; sample concentration: 10 mg/mL. Molecular mass standards include bacitracin, aprotinin, myohemoglobin, bovine serum albumin, and thyroglobulin. The molecular weight of the protein was determined using the regression equation for molecular weight. Furthermore, the standard curve was plotted with the retention time of the standard as the abscissa, and log (MW) as the ordinate to obtain the regression equation.

#### 2.3.3. Sodium Dodecyl Sulfate-Polyacrylamide Gel Electrophoresis (SDS-PAGE)

SDS-PAGE images of SPI, PSPI, and ASPI were analyzed by SDS-PAGE according to the method of Jiang et al. [21]. The stacking and separating gels contained 4% and 12% acrylamide, respectively. Equal volumes of samples were mixed with loading buffer in the presence and absence of *β*-mercaptoethanol to a concentration of 2 mg/mL, and the mixtures were boiled for 5 min at 100 °C. A standard medium molecular weight protein was used as a molecular weight marker. Each lane was loaded with 15 μL of the sample, and the initial voltage was set to 80 V. The voltage was increased to 120 V once the samples reached the separating gel. Staining with BeyoBlue™ Coomassie Blue Super Fast Staining Solution and destaining with distilled water was performed prior to scanning the gel image with Bio-Rad’s Image Capture System.

#### 2.3.4. Surface Hydrophobicity Determination

Protein surface hydrophobicity (H_0_) was determined using the ANS probe assay [22]. All ANS (8 mM) and protein samples (0.2–1 mg/mL) were prepared in pH 7.0 phosphate buffer (10 mM). The 4 mL protein samples were thoroughly mixed with 20 μL ANS. After 3 min of reaction in the dark, the fluorescence intensity was measured using an F-2700 spectrofluorometer (Hitachi, Tokyo, Japan) under the conditions described below. Excitation wavelength: 390 nm; emission wavelength: 470 nm; slit width: 5 nm. The standard curve was plotted with the protein concentration (mg/mL) of samples as the abscissa and the fluorescence as the ordinate, and the slope of the curve was determined as H_0_.

### 2.4. Preparation of Lutein-Loaded Nanoparticles

The lutein-loaded nanoparticles of SPI (SPI-LNPS), PSPI (PSPI-LNPS), and ASPI (ASPI-LNPS) were fabricated using the method described by Chen et al. [23] with some modifications. Protein (2.5 mg/mL) was dissolved using a 10 mM phosphate buffer (pH 7.0) with complete hydration at 4 °C. Lutein solutions (0.25, 0.5, 0.75, and 1 mg/mL) in absolute ethyl alcohol were prepared using a two-min sonication treatment with 5 s of sonication and 5 s of rest. At a ratio of 1:10, lutein solutions of various concentrations were injected into protein solutions (*v*/*v*). After 2.5 h of magnetic stirring, the insoluble lutein was removed via 15 min of centrifugation at 10,000 g and determined as free lutein. The supernatant obtained was used as lutein-loaded nanoparticles.

### 2.5. Encapsulation Efficiency (EE) and Loading Capacity (LC) of Lutein

EE and LC were determined in accordance with Yuan et al. [15]. Ethanol was used to extract the encapsulated lutein. The previously obtained supernatants were combined with ethanol to achieve a final ethanol concentration of 90%. Following complete protein precipitation, the mixtures were centrifuged at 10,000× *g* for 20 min. The supernatants were collected and, if necessary, diluted with 90% ethanol. The encapsulated lutein was then quantified using spectrophotometric analysis at 445 nm against a known lutein curve (R^2^ > 0.999). The EE and LC of lutein were determined using the following Equations (2) and (3):EE (%) = Encapsulated lutein (mg) / Lutein input (mg) × 100(2)
LC = Encapsulated lutein (μg) / Total mass of protein (mg)(3)

### 2.6. Particle Characteristics

#### 2.6.1. Particle Size, Polydispersity Index (PDI) and ζ-Potential

The particle size, PDI, and ζ-potential of protein solutions and nanoparticles were determined using a Zetasizer Nano-ZS (Malvern Instruments, Malvern, UK) with reference to the method of Wu et al. [24]. The fixed angle, refractive index, and test temperature were all set to 173, 1.33, and 25 °C, respectively.

#### 2.6.2. Atomic Force Microscopy (AFM) Images

AFM images were obtained according to the method of Feng et al. [25]. AFM images were captured in the air using the tapping mode with an SNL-10 probe on a Dimension Icon scanning probe microscope (Bruker Technology Co., Ltd., Bremen, Germany). The scan rate was set to 256 Hz. Protein and lutein-loaded nanoparticles were diluted to a protein concentration of 5 μg/mL by deionized water and 2 μL was added to the surface of a freshly peeled mica sheet. The samples were dried overnight in the air before being captured under the microscope. Nanoscope Analysis 1.9 software (Bruker Corp., Santa Barbara, CA, USA), was used to further process the obtained images.

### 2.7. Light Stability

The stability of light was investigated using the method described by Du et al. [17]. Unencapsulated lutein was dispersed in a 10 mM phosphate buffer (pH 7.0) using ultrasound, and nanoparticles loaded with an equivalent amount of lutein were exposed to 15,000 LX white light at 37 °C for 48 h using a GZP-150N illumination incubator (Senxin Instruments Ltd., Shanghai, China). The amount of lutein retained was determined using the method described previously (Section 2.5). Furthermore, to represent light stability, the retention rate of lutein was plotted against time.

### 2.8. In Vitro Gastrointestinal Digestion

The in vitro digestive stability of unencapsulated and encapsulated lutein was determined using the method described by Li et al. [26]. In equal proportions, unencapsulated lutein, SPI-LNPS, and PSPI-LNPS were combined with simulated gastric fluid (SGF). The mixtures (pH 3.0) were shaken at 200 rpm for 2 h at 37 °C to simulate stomach digestion. The mixtures were then mixed with simulated intestinal fluids (SIF) in equal parts. Then the mixtures (pH 7.0) were shaken for 2 h to simulate intestinal digestion. With reference to Jiao et al. [8], 0.5 mL suspensions were taken and extracted with 2.5 mL of ethanol. After thorough mixing and complete protein precipitation, the mixtures were centrifuged at 20,000× *g* for 20 min, and the supernatants were analyzed for retained lutein content using an HPLC system (Waters 2695, Milford, MA, USA) equipped with a 2487 UV detector (Waters Corp., Milford, MA, USA) operating at 445 nm.

In addition, the following conditions were used during testing: chromatography column: XBridge C18 column (4.6 × 250 mm, 5 μm, Waters Corp., Milford, MA, USA); wavelength: 445 nm; flow rate: 1 mL/min; column temperature: 30 °C; injection volume: 10 μL; mobile phase: (A) absolute methanol and (B) 0.1% methanoic acid with linear gradient elution (0–20 min, 90%–100% A; 20–25 min, 100%–90% A).

### 2.9. Measurement of Intrinsic Fluorescence

This section was implemented according to the method described by Ma et al. [9]. Protein solutions (0.4 mg/mL) of SPI and PSPI were prepared in a 10 mM phosphate buffer (pH 7.0). As a stock solution, lutein was dissolved in absolute ethanol at a concentration of 0.1 mM. Subsequently, to achieve a final lutein concentration of 0−30 μM, various concentrations of lutein stock solution were added to protein solutions with complete mixing. At 34 °C, the intrinsic fluorescence was determined using an F-2700 fluorescence spectrometer (Hitachi Ltd., Tokyo, Japan). The following parameters were used during the test: excitation wavelength 280 nm, emission wavelength 300–450 nm, slit width 5.0 nm, and scan speed 1500 nm/min.

The Stern-Volmer equation was used to investigate the fluorescence quenching process of protein, in order to clarify fluorescence quenching mechanisms.
F_0_/F = 1 + K_sv_[Q] = 1 + k_q_τ_0_[Q](4)
where F_0_ is the fluorescence intensity before lutein addition; F is the fluorescence intensity after lutein addition; [Q] is the final lutein concentration (μM); K_sv_ denotes Stern-Volmer quenching; k_q_ denotes the quenching rate constant; and τ_0_ denotes the fluorescence lifetime of the fluorophore in the absence of quenching, which is typically 10^−8^ s.

The value of k_q_ confirms the quenching type. The binding constant (K_a_) and the number of binding sites (n) for static quenching can be determined using the Stern-Volmer logarithmic equation as follows:log(F_0_−F) / F = logK_a_ + nlog[Q](5)

### 2.10. Circular Dichroism (Cd) Analysis

Secondary structures of SPI, PSPI, and their mixtures with lutein were determined using the method of Shen et al. [19] on a Chirascan V100 CD Spectrometer (Applied Photophysics Ltd., Leatherhead, UK). Protein and lutein concentrations were adjusted to 0.1 mg/mL and 0.03 mg/mL, respectively. The CD spectral range was scanned between 190 and 260 nm. Additionally, the following parameters were demonstrated: a scanning speed of 60 nm/min; spectral resolution of 0.2 nm; a response time of 0.25 s; and a slit width of 1 nm. The CDPro software was used to determine the proportion of secondary structure (Applied Photophysics).

### 2.11. The Fourier Transform Infrared Spectra (FTIR) and X-ray Diffraction (XRD) Analysis

The FT-IR and XRD spectra were obtained using the method described by Li et al. [26]. The FTIR spectra were recorded using a Nicolet IS10 spectrometer (Thermo Fisher Inc., Waltham, MA, USA). Lyophilized lutein, SPI, PSPI, and lutein-loaded nanoparticles were fully blended at a ratio of 1:100 (*w*/*w*) with freshly dried KBr powder and compressed to a pellet for measurement. The scanning wavelength was adjusted to be between 500 and 4000 cm^−1^.

A D2 Phaser diffractometer was used to obtain XRD spectra (Bruker). The 2θ angle was adjustable between 5° and 50° with a step size of 0.05°.

### 2.12. Statistical Analysis

The experiments were repeated three times, and data were presented as the mean ± standard deviation (SD). Statistix 9.0 software was used to conduct the statistical analysis (Analytical Software, Tallahassee, FL, USA). Significant differences were determined by the least significance difference (LSD) method. When *p* < 0.05, a statistical difference existed.

## 3. Results and Discussion

### 3.1. Characteristics of SPI, PSPI, and ASPI

Enzymatic hydrolysis of food-derived protein for bioactive compound nanoencapsulation has been widely reported previously [8,17]. Different DH values for proteins resulted in varying degrees of encapsulation [9].

PSPI and ASPI had a DH of 6.00% and 14.96%, respectively (Table 1), indicating that alcalase had a greater capacity for hydrolysis of SPI than pepsin. Consistent with the result of DH, the MW distribution and nonreducing SDS-PAGE analysis of SPI and its hydrolysates revealed a similar trend.

As illustrated in Figure 1a, the larger peptides with an MW of >400 kDa of SPI were predominantly hydrolyzed to peptides with an MW of 10–100 kDa by pepsin and alcalase. Similar results were found in previous research that the proportion of peptides with low MW for SPI increased after hydrolysis by alcalase and pepsin [24,27]. In comparison to PSPI, the proportion of peptides with an MW of 10–50 kDa was approximately 17% higher in ASPI. As shown in Figure 1b, the majority of MW bands in ASPI were less than 31 kDa on the nonreducing gel. However, PSPI contained MW bands between 40 and 100 kDa. According to reducing SDS-PAGE, *α*, *α*′, and *β* subunits of *β*-conglycinin remained in PSPI, whereas the A and B subunits of glycinin were mostly hydrolyzed, which was consistent with the findings of Chen et al. [28]. The hydrophobicity of *β*-conglycinin was found to be greater than that of glycinin [29], which may explain why the H_0_ of PSPI, with a higher proportion of *β*-conglycinin subunits, was greater than that of ASPI with no glycinin or *β*-conglycinin subunits remaining (Table 1). The differences in the ASPI and PSPI characteristics may be due to the different enzymatic hydrolysis properties of pepsin and alcalase on SPI. Alcalase possesses broad specificity endoprotease activity, which enables it to obtain a DH value greater than 10% after a brief period of hydrolysis, resulting in a relatively low H_0_ [30]. Pepsin is an aspartic protease that can hydrolyze glycinin but retain its *β*-conglycinin of SPI [31].

### 3.2. Encapsulation Properties and Particle Characteristics

According to Figure 2a,b, as the amount of lutein added increased, the EE of lutein in PSPI-LNPS and ASPI-LNPS decreased, whereas the EE of lutein in SPI-LNPS reached the highest at the mass ratio of 50:1. The LC of lutein increased and then decreased in SPI-LNPS and PSPI-LNPS, whereas the LC of lutein in ASPI-LNPS decreased with lutein addition. It is worth noting that when the lutein was added at a mass ratio of 100:3, the LC of lutein in PSPI-LNPS was the highest (17.96 μg/mg protein), approximately 2.28%, and eight times higher than that of SPI-LNPS and ASPI-LNPS, respectively, and the corresponding lutein EE was 5.83%, and 10 times higher than that of the other two nanoparticles. So the mass ratio of 100:3 was chosen for further index determination. It was concluded that PSPI was the most effective at encapsulating lutein at a relatively high lutein loading level, whereas ASPI was the least effective. According to previous research, hydrophobic bonding was the primary mechanism by which hydrophobic bioactives were encapsulated [8], which explains why the EE and LC values of lutein in SPI, PSPI, and ASPI were positively correlated with their H_0_ values in this study (Table 1).

For particle characteristics shown in Figure 2c,d, PSPI had a single peak distribution, whereas SPI and ASPI had two and three peaks, respectively. This finding was consistent with the data in Table 2, which indicated that the PDI of PSPI was <0.3. The addition of lutein increased the particle sizes of SPI-LNPS, PSPI-LNPS, and ASPI-LNPS, which was consistent with the previous research by Qi et al. [32]. It was discovered that the addition of lutein dipalmitate increased the size of BSA, which could be related to the interaction between BSA and lutein dipalmitate. However, the size distribution and PDI of nanoparticle trends were similar to that of pure protein, with a single peak and the lowest PDI of 0.11 in PSPI-LNPS. PSPI-LNPS was chosen for further investigation due to its superior encapsulation properties, size distribution, and PDI.

### 3.3. Effect of Light and Digestive Environment on the Stability of Lutein-Loaded Nanoparticles

The lutein retention rate was determined to investigate the stability of lutein in SPI-LNPS, PSPI-LNPS, and unencapsulated lutein when exposed to strong light and gastrointestinal digestion in vitro.

As illustrated in Figure 3a, lutein in SPI-LNPS showed rapid degradation after 48 h of light exposure, which contrasted with the previous research showing that *β*-carotene’s light stability was significantly improved when encapsulated in *α*-Lactalbumin [17]. This may be due to the nonuniformity of particles with large aggregates present in SPI-LNPS, as depicted by the size distribution results in Figure 2d, indicating that SPI-LNPS may be easily degraded when exposed to intermediate products derived from lutein autoxidation [33]. After 48 h of light exposure, lutein retention in PSPI-LNPS was 85.05%, which was approximately 11.13% higher than that of unencapsulated lutein. This result may also be reflected in the fact that PSPI-LNPS has a darker color than unencapsulated lutein and SPI-LNPS in Figure 3a.

As illustrated in Figure 3b, the retention rate of lutein in SPI-LNPS was found to be 5.32% greater than that of unencapsulated lutein following in vitro gastric digestion (*p* < 0.05). When lutein was encapsulated with PSPI, the retention rate increased by 5.52% (*p* < 0.05), when compared with SPI-LNPS. After intestinal digestion, SPI-LNPS retained lutein at a non-significantly higher rate (*p* > 0.05) than unencapsulated lutein, whereas PSPI-LNPS retained lutein at a significantly higher rate with an improvement of approximately 8.40% (*p* < 0.05). Jiao et al. [8] also reported that encapsulating lutein via zein-derived peptides significantly reduced lutein degradation in SGF and SIF. These findings indicated that encapsulating lutein with SPI hydrolysates via pepsin significantly improved lutein stability in the presence of light and a simulated digestive environment.

### 3.4. Fluorescence Spectroscopy Analysis

In order to understand why PSPI-LNPS had a better encapsulation performance and stability against light and in vitro digestion, fluorescence emission spectroscopy was used to analyze the interaction between protein and lutein. Due to the presence of tryptophan (Trp), tyrosine (Tyr), and phenylalanine (Phe) residues in proteins, endogenous fluorescence can occur [34]. The fluorescence intensity towards wavelength changed in a diverse solvent environment.

In Figure 4a,b, SPI emitted at a high fluorescence intensity of 339 nm, whereas a redshift to the highest fluorescence emission wavelength of 350 nm happened in PSPI, indicating a more open structure of the protein caused by the demasking of peptide bonds following hydrolysis [35].

The fluorescence intensity of SPI and PSPI decreased as lutein concentration was increased (Figure 4a,b), indicating that the interaction between protein and lutein quenched the protein’s intrinsic fluorescence. Meanwhile, the addition of lutein resulted in a slight blueshift of about 3 nm, indicating that the aromatic residues were shifted to a more hydrophobic environment [15].

As illustrated in Figure 4a,b, a strong linear relationship between F_0_/F and lutein concentration was observed in both SPI and PSPI. The Stern-Volmer equation was used to calculate the quenching constant K_sv_ and the quenching rate constant k_q_. As the values of k_q_ in SPI-LNPS and PSPI-LNPS were significantly greater than 2.0 × 10^10^ M^−1^ s^−1^, the quenching mechanism was static [32]. In order to further compare the binding capacity of lutein between SPI and PSPI, the Stern-Volmer logarithmic equation was used to determine the binding constant (K_a_) and the number of binding sites (n).

In Table 3, the values of K_a_ and n for lutein-PSPI were both greater than those for lutein-SPI, indicating that lutein had a greater binding capacity for PSPI than SPI. The stronger interaction between lutein and PSPI was positively associated with lutein’s increased loading capacity and increased stability against light exposure and a simulated digestive environment in PSPI-LNPS. This result was also consistent with a previous work of Ma et al. [9] on lutein encapsulation using zein-derived peptides. The difference in lutein binding capacity between SPI and PSPI was most likely caused by the fact that PSPI exposed more hydrophobic binding sites to the surface of the protein following hydrolysis with pepsin with a higher H_0_ than SPI (Table 1).

### 3.5. CD Analysis

The secondary structure of SPI and PSPI was determined by CD spectra. The mean ellipticity of the residues between 206 and 220 nm indicates the presence of α-helixes in the protein [36]. As shown in Figure 5, SPI exhibited a positive peak around 196 nm and a broad negative peak at 206 nm. The positive peak vanished after pepsin hydrolysis, while the negative peak blue shifted to 202 nm with a larger negative value. This result was consistent with previous research using alcalase to hydrolyze SPI [24]. The ratio of *α*-Helix to *β*-Sheet can be used to demonstrate the flexibility and stiffness of proteins [19].

According to Table 4, the proportion of *α*-Helix in SPI decreased and was converted to a random coil following pepsin hydrolysis, indicating that the increased flexibility in PSPI was consistent with the fluorescence spectrum analysis result. This phenomenon was also reflected by a decrease in the mean residue ellipticity at approximately 206 nm, which corresponds to the *α*-Helix content. With the addition of lutein, the secondary structure composition of SPI and PSPI remained nearly unchanged, as previously demonstrated in a study on the interaction of proanthocyanidins and soybean seed ferritin [37].

### 3.6. FT-IR and XRD Analysis

The total composition of biomaterials was determined using FT-IR. According to Figure 6a, lutein exhibited distinct peaks at 2956, 2917, 2850, 1600, 1157, 1042, and 964 cm^−1^. Three of these peaks were previously identified by Jiao et al. [8]. Three peaks at 1157, 1066, and 952 cm^−1^, which did not exist in SPI or PSPI, were detected in SPI-LNPS and PSPI-LNPS, respectively. This phenomenon was also observed in a previous study conducted by Feng et al. [38], which demonstrated the successful encapsulation of lutein in nanoparticles. Additionally, FT-IR can provide information about the structure of proteins. The Amide I band between 1600 and 1700 cm^−1^ is attributed to the C=O stretch and can be used to analyze protein secondary structure. Between 1500 and 1600 cm^−1^, the Amide II band indicates the vibrations of N–H bending and C–N stretching. As shown in Figure 6a, both the Amide I and II bands of PSPI showed a redshift when compared to SPI, indicating that the structure of SPI changed following hydrolysis. The Amide I and II bands in SPI-LNPS and PSPI-LNPS seldom shifted when lutein was added, indicating that the protein structure remained nearly unchanged, consistent with their CD results.

XRD was used to confirm the lutein’s physical state (Figure 6b). In unencapsulated lutein, strong crystal diffraction peaks with two thetas ranging from 5° to 25° were observed. SPI revealed two broad weak peaks at 9.1° and 20.5°. PSPI, SPI-LNPS, and PSPI-LNPS exhibited almost no diffraction peaks. Similarly, when lutein was encapsulated in glycosylated casein, similar results were obtained [7]. This result confirmed that the protein and nanoparticles were amorphous [39], indicating that the lutein was encapsulated in the protein’s hydrophobic core, which was more advantageous for the delivery system than unencapsulated lutein [40].

### 3.7. Morphology Analysis

The morphology of the protein and lutein-loaded nanoparticles depicted in Figure 7 was consistent with the particle size determination result. SPI contained some large aggregated particles (Figure 7a), whereas PSPI contained particles that were uniformly distributed and had similar particle sizes (Figure 7c). Increased particle size was also observed with the addition of lutein in both SPI-LNPS (Figure 7b) and PSPI-LNPS (Figure 7d), which was due to the interaction of lutein and protein [32]. These findings strongly suggest that hydrolysis of SPI with pepsin can result in the formation of more uniform spherical nanoparticles.

## 4. Conclusions

In this study, SPI and its hydrolysates via pepsin and alcalase (PSPI, ASPI) were used to encapsulate lutein. Compared to SPI-LNPS, PSPI-LNPS demonstrated increased lutein loading capacity, uniform particle size distribution, and improved lutein stability against light exposure and simulated gastric environment, whereas ASPI-LNPS demonstrated the poorest encapsulation properties and particle characteristics, which were positively related to the *β*-conglycinin content and surface hydrophobicity of the protein. Fluorescence spectroscopy and CD spectra revealed that after pepsin hydrolysis, the SPI structure became more open and flexible, and the PSPI exhibited a greater binding strength for lutein than the SPI, primarily due to hydrophobic interaction, thereby contributing to lutein’s improved encapsulation property, light and digestive stability in PSPI-LNPS. These findings indicate that certain SPI-derived peptides have the potential to significantly increase the stability of lutein and be used as a nanoencapsulation material for lutein-enriched nutraceuticals.

## Figures and Tables

**Figure 1 foods-11-03635-f001:**
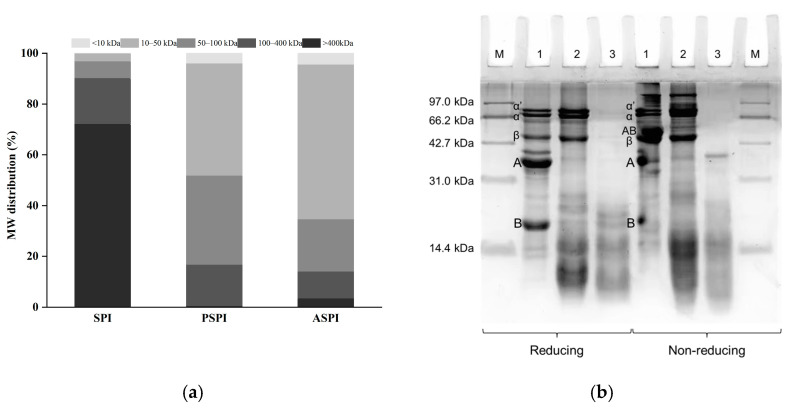
SPI, PSPI, and ASPI structural characteristics. (**a**) SPI, PSPI, and ASPI molecular weight distribution; (**b**) SDS-PAGE profiles of SPI (Lane 1), PSPI (Lane 2), ASPI (Lane 3), and a molecular mass standard (Lane M).

**Figure 2 foods-11-03635-f002:**
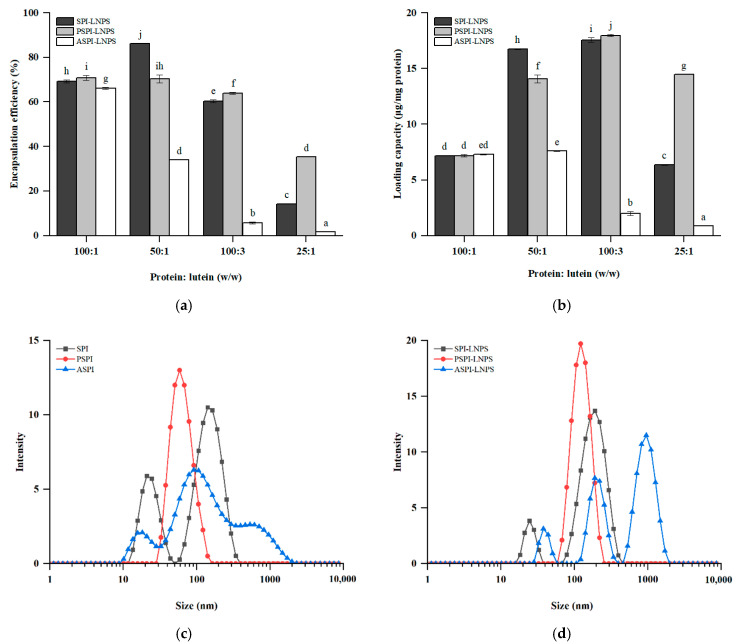
(**a**) Encapsulation efficiency (EE) of lutein in SPI-LNPS, PSPI-LNPS, and ASPI-LNPS; (**b**) loading capacity (LC) of lutein in SPI-LNPS, PSPI-LNPS, and ASPI-LNPS; (**c**) SPI, PSPI, and ASPI particle size distributions under protein concentration of 0.25 mg/mL; (**d**) SPI-LNPS, PSPI-LNPS, and ASPI-LNPS particle size distributions under protein concentration of 0.25 mg/mL with the mass ratio of 100:3 between protein and lutein. The standard deviation is represented by the bars. Columns denoted by different letters (a–j) indicate a statistically significant difference (*p* < 0.05).

**Figure 3 foods-11-03635-f003:**
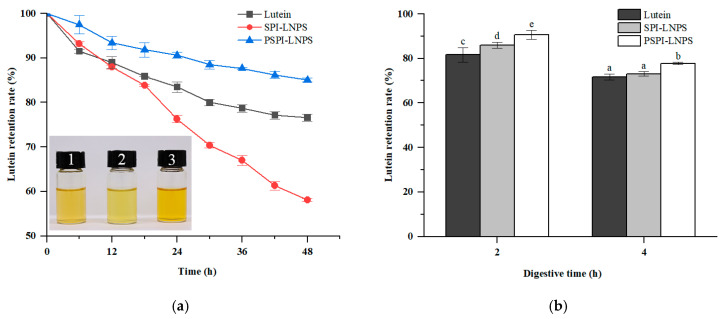
(**a**) Lutein retention rate of unencapsulated lutein, SPI-LNPS, and PSPI-LNPS after 48 h of light exposure; (**b**) Lutein retention rate of unencapsulated lutein, SPI-LNPS, and PSPI-LNPS after 4 h of digestion. Lutein (1), SPI-LNPS (2), and PSPI-LNPS (3) were photographed in (**a**) after 48 h of exposure to light. The standard deviation is represented by the bars. Columns with different letters (a–e) indicate a significant difference between samples (*p* < 0.05).

**Figure 4 foods-11-03635-f004:**
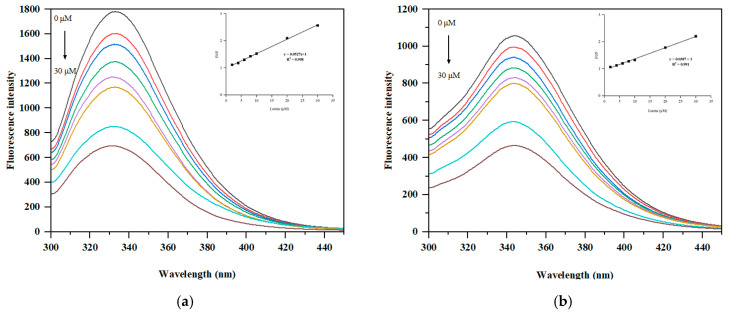
(**a**) Fluorescence emission spectra of SPI; (**b**) Fluorescence emission spectra of PSPI. protein concentration was 0.4 mg/mL and the lutein additive amount was 0, 2, 4, 6, 8, 10, 20, and 30 μM.

**Figure 5 foods-11-03635-f005:**
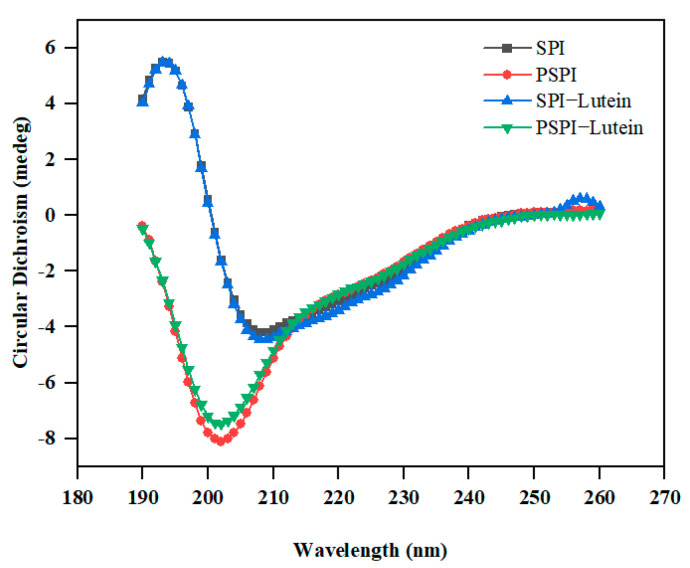
CD spectra of SPI and PSPI (0.1 mg/mL) in the absence and presence of lutein (0.03 mg/mL).

**Figure 6 foods-11-03635-f006:**
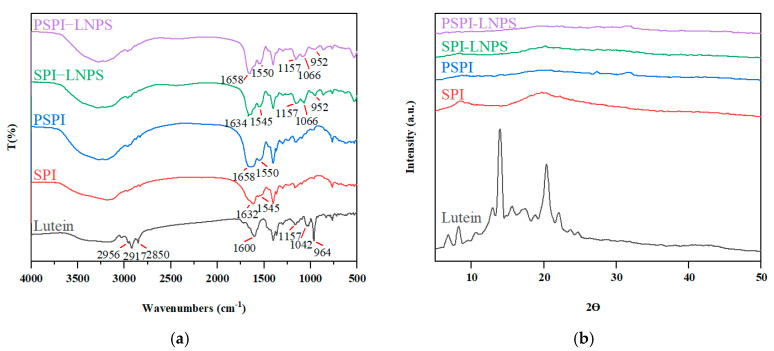
(**a**) FTIR spectra of lutein, SPI, and PSPI (2.5 mg/mL) in the absence and presence of lutein (0.07 mg/mL); (**b**) XRD spectra of lutein, SPI, and PSPI (2.5 mg/mL) in the absence and presence of lutein (0.07 mg/mL).

**Figure 7 foods-11-03635-f007:**
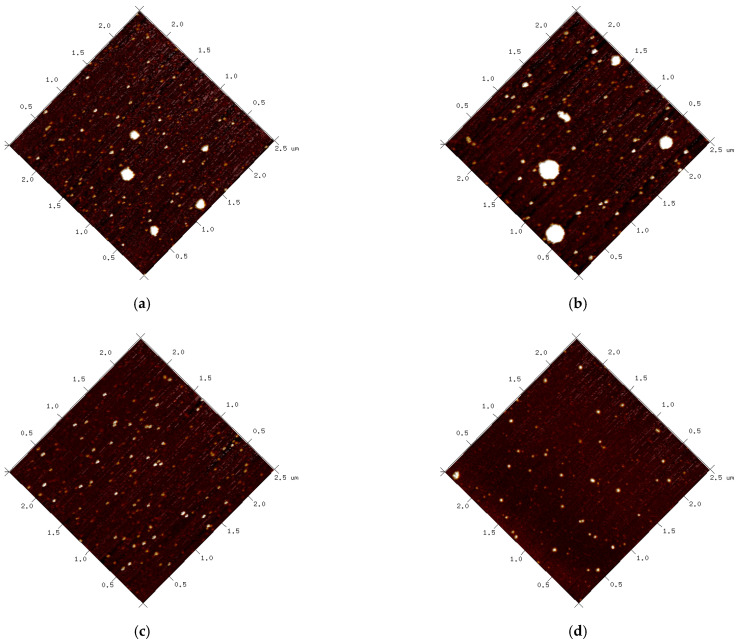
AFM images. (**a**) SPI; (**b**) SPI-LNPS at a mass ratio of 100:3 between protein and lutein; (**c**) PSPI; (**d**) PSPI-LNPS at a mass ratio of 100:3 between protein and lutein.

**Table 1 foods-11-03635-t001:** Degree of hydrolysis (DH) and surface hydrophobicity (H_0_) of SPI, PSPI, and ASPI.

Sample	DH (%)	H_0_
SPI	-	86.24 ± 0.51 ^b^
PSPI	6.00 ± 0.05 ^a^	140.27 ± 1.76 ^c^
ASPI	14.96 ± 0.04 ^b^	20.78 ± 0.12 ^a^

Values are expressed as the mean ± SD. Different letters (^a^, ^b^, and ^c^) denote a significant difference (*p* < 0.05) between samples in the same column.

**Table 2 foods-11-03635-t002:** Average particle size, PDI, and ζ-potential of SPI, PSPI, and ASPI and their lutein-loaded nanoparticles.

Sample	Average Particle Size (nm)	PDI	ζ-Potential (mV)
SPI	59.5 ± 0.34 ^a^	0.55 ± 0.005 ^e^	−25.30 ± 0.99 ^c^
PSPI	71.12 ± 0.37 ^b^	0.26 ± 0.003 ^b^	−20.50 ± 0.71 ^a^
ASPI	76.23 ± 5.17 ^b^	0.52 ± 0.03 ^d^	−23.10 ± 0.57 ^b^
SPI–LNPS	124.43 ± 1.85 ^d^	0.48 ± 0.01 ^c^	−23.25 ± 0.92 ^b^
PSPI–LNPS	114.47 ± 1.45 ^c^	0.11 ± 0.009 ^a^	−23.45 ± 0.78 ^b^
ASPI–LNPS	236.10 ± 8.20 ^e^	1.00 ± 0.00 ^f^	−25.35 ± 0.07 ^c^

Values are expressed as the mean ± SD. Different letters (^a–f^) denote a significant difference (*p* < 0.05) between samples in the same column.

**Table 3 foods-11-03635-t003:** The quenching constant (K_sv_), the quenching rate constant (k_q_), the binding constant (K_a_), and the number of binding sites (n) for Lutein (0–30 μM) binding to SPI and PSPI.

	k_q_/10^12^ M^−1^ s^−1^	K_sv_/10^4^ M^−1^	K_a_/10^5^ M^−1^	n
Lut-SPI	5.27 ± 0.12 ^b^	5.27 ± 0.12 ^b^	0.63 ± 0.03 ^a^	1.02 ± 0.01 ^a^
Lut-PSPI	3.87 ± 0.17 ^a^	3.87 ± 0.17 ^a^	1.04 ± 0.12 ^b^	1.10 ± 0.03 ^b^

Values are expressed as the mean ± SD. Different letters (^a^ and ^b^) denote a significant difference (*p* < 0.05) between samples in the same column.

**Table 4 foods-11-03635-t004:** The secondary structure content evaluated by CDPro of SPI and PSPI with and without the addition of lutein.

Secondary Structure	*α*-Helix (%)	*β*-Sheet (%)	*β*-Turns (%)	Random Coil (%)
SPI	16.24 ± 0.08 ^b^	32.84 ± 0.47 ^a^	22.24 ± 0.06 ^c^	28.69 ± 0.45 ^b^
PSPI	9.27 ± 0.05 ^c^	31.67 ± 0.27 ^b^	23.00 ± 0.02 ^a^	36.07 ± 0.28 ^a^
SPI-Lut	16.94 ± 0.51 ^a^	31.67 ± 0.68 ^b^	22.31 ± 0.09 ^c^	29.08 ± 0.11 ^b^
PSPI-Lut	8.74 ± 0.25 ^c^	32.92 ± 0.34 ^a^	22.78 ± 0.15 ^b^	35.56 ± 0.40 ^a^

Values are expressed as the mean ± SD. Different letters (^a–c^) denote a significant difference (*p* < 0.05) between samples in the same column.

## Data Availability

The data presented in this study are available on request from the corresponding author.

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
