# Peer review of "Improved Light and In Vitro Digestive Stability of Lutein-Loaded Nanoparticles Based on Soy Protein Hydrolysates via Pepsin"

_foods, 2022, doi:10.3390/foods11223635_

Round 1
Reviewer 1 Report
The topic that has been presented in the manuscript is interesting. I have some minor suggestions
the introduction can be improved in a way to focus on the similar approaches. Concentrating too much on the several agents for encapsulation is misleading.
also in the last paragraph of the introduction, the auther has mentioned the in vitro study which is not presented here in the data.
Reviewer 2 Report
Authors have demonstrated light and digestive stability of lutein-loaded nanoparticles. Overall, the manuscript has some merits. However, there are some key information that are missing and authors are advised to revise or clear the following queries;
1. The manuscript requires English language proofreading.
2. On page 3, line no. 104, Authors mentioned enzymatic hydrolysis took 45 minutes at two different conditions (pepsin at 2.0 pH) and (alcalase at pH 8.0); however, it is not mentioned how much time was given for each condition.
3. On page 4, line no.161, authors mention 2 min. sonication. Please elaborate this further. At what frequency sonication was performed?
4. In Section 2.5, please recheck equation 2 and 3 for EE% and LC calculations.
5. Which statistical method was used to check the significance in difference? Please mention.
